# Effects of Sevoflurane on Lewis Lung Carcinoma Cell Proliferation In Vivo and In Vitro

**DOI:** 10.3390/medicina57010045

**Published:** 2021-01-07

**Authors:** Yeojung Kim, Sangwon Yun, Keun-A Shin, Woosuk Chung, Youngkwon Ko, Yoon-Hee Kim, Chaeseong Lim

**Affiliations:** 1Department of Anesthesiology and Pain Medicine, Chungnam National University Sejong Hospital, Sejong 30099, Korea; yeojung80@naver.com; 2Department of Anesthesiology and Pain Medicine, Chungnam National University College of Medicine, Daejeon 35015, Korea; woosuk119@gmail.com (W.C.); annn8432@gmail.com (Y.K.); yhkim040404@gmail.com (Y.-H.K.); 3Department of Anesthesiology and Pain Medicine, Chungnam National University Hospital, Daejeon 35015, Korea; yfreedom03@naver.com; 4Research Institute for Medical Sciences, Chungnam National University College of Medicine, Daejeon 35015, Korea; smmra98@naver.com

**Keywords:** sevoflurane, Lewis lung carcinoma, cancer, anesthetics, survival

## Abstract

*Background and objectives:* There are several studies that sevoflurane could enhance proliferation of cancer cells, while others suggest no effect on clinical outcome. We conducted in vivo and in vitro experiments to investigate the effects of sevoflurane, a volatile anesthetic, on proliferation and outcomes of Lewis lung carcinoma (LLC) cells. *Materials and Methods:* A total of 37 mice were injected with LLC cells to compare the tumor size and survival of the sevoflurane exposed group (sevo group) and control group. The sevo group was exposed to 2% sevoflurane and 4 L/min of oxygen for 1 h per day 3 times per week, and the control group was exposed only to 4 L/min of oxygen. In vitro study, 12 plates incubated with LCC cells. 6 plates were exposed to 2% sevoflurane for 1 hr/day for 3 days and 6 plates were not exposed, and cell proliferation was compared after 3 days. *Results:* There were no significant differences in survival or tumor size between mice exposed to sevoflurane and control mice (survival: 29.06 ± 4.45 vs. 28.76 ± 3.75, *p* = 0.836; tumor size: 0.75 (0.41–1.02) vs. 0.49 (0.11–0.79), *p* = 0.153). However, in vitro study, the proliferation of LLC cells exposed to sevoflurane increased by 9.2% compared to the control group (*p* = 0.018). *Conclusions:* Sevoflurane (2 vol%) exposure could promote proliferation of LLC cells in vitro environment, but may not affect proliferation of LLC cells in vivo environment. These results suggest that in vitro studies on the effects of anesthetics on cancer may differ from those of in vivo or clinical studies.

## 1. Introduction

Cancer is the second most common cause of death worldwide [1,2]. In developed countries, cancer is already the most common cause of death and also expected that cancer is able to convert the main cause of death universally within several years [3]. Cancers are treated with a combination of therapies (surgery, radiation, chemotherapy and etc.) depending on the stage and type of cancer. According to recent reports, more than 60% of cancer patients require surgery to remove solid tumors among above treatments [4].

There may be several factors adversely affect cancer outcome in perioperative periods [5,6]. Perioperative immunocompromise, pain and opioids could affect proliferation and invasion of residual cancer cells [7,8,9]. In addition, there are several factors can cause impaired local barrier and cell mediated immunity in perioperative period, resulting local recurrence and metastasis. The factors are such as manipulation of the tumor, physical tissue damage, and stresses to the physiologic stimulations [2,9,10,11,12]. It has been suggested that anesthetic technique could affect these factors [10,13].

Anesthetic methods have a variety of effects on enhancement of pro-tumorigenic cellular signaling, and humoral and cell mediated immunity. These influences could be comprehended an impact of type of anesthetics on long-term prognosis of cancer [2,14]. Reviews of current clinical practice describe that most of anesthesiologists commonly prefer volatile anesthetic agents such as the sevoflurane [2].

There are several clinical trials on effects of volatile anesthetic agent on cancer cell proliferation, but they are mostly retrospective studies, and a few RCTs [2,14,15,16,17,18,19]. That reports were evaluated only indirect effects. There are also several in vitro studies [8,13,20,21,22,23,24,25,26], but a few in vivo studies [27]. Direct tumor proliferation and outcome can be compared through an in vivo study.

In the current studies, we investigate the effects of sevoflurane, a volatile anesthetic, on the outcomes and proliferation of cancer by injecting Lewis lung carcinoma (LCC) cells into the buttocks of mice and comparing tumor size and life span of the sevoflurane exposed group and sevoflurane unexposed groups in vivo. In addition, we compare the proliferation of Lewis carcinoma cells exposed to sevoflurane with them unexposed to sevoflurane in vitro.

## 2. Materials and Methods

### 2.1. In Vivo Study

The animal protocol was reviewed and approved by the Institutional Animal Care and Use Committee of the Chungnam National University, Daejeon, Korea (No. CNU-01178, 26 December 2018). To assess the in vivo effects of sevoflurane exposure on cancer cells, we used an LLC xenograft mouse model. As a primary endpoint, mouse survival was determined from the day of LLC cell injection (day 0), while the tumor size on day 18 was used as the secondary endpoint. A schematic illustration of the procedures followed in the in vivo study is shown in Figure 1.

#### 2.1.1. Cell Culture

LLC cells were obtained from the American Type Culture Collection (ATCC). They were maintained in Dulbecco’s modified Eagle medium (DMEM) supplemented with 10% fetal bovine serum (FBS), 100 U/mL penicillin, and 100 μg/mL streptomycin at 37 °C in a humidified 5% CO_2_ atmosphere.

#### 2.1.2. Animals

Male C57BL/6J mice (6 weeks old) were allowed to acclimate in our facility for 7 days before use. They were given ad libitum access to food and water and caged in groups of four.

#### 2.1.3. Mouse Xenograft Model and Sevoflurane Exposure

To establish a mouse xenograft model, we prepared 0.1 mL phosphate-buffered saline (PBS; pH, 7.0) solution containing 1 × 10^6^ LLC cells. The cell solution was injected subcutaneously into the right flank of each mouse.

The day after the injection, the animals were sited in a 2 L airtight gas container (4 in each container) with inlet and outlet valves, in a 37 °C water bath. For sevoflurane delivery, the container inlet was connected to a closed gas delivery system containing a calibrated oxygen flow meter and a sevoflurane vaporizer (Abbott Laboratories, Maidenhead, UK). Outlet gas was monitored (Datex gas monitor, Helsinki, Finland) until a sevoflurane concentration of 2.0% was reached in 4 L/min flow oxygen; sevoflurane delivery was performed for 1 h three times per week until death (sevo group). Control mice were placed in the container with 4 L/min oxygen flow without sevoflurane; oxygen delivery was performed for 1 h three times per week until death.

#### 2.1.4. Tumor Size and Mouse Survival

Survival was recorded from the day after LLC cell xenograft. Tumor sizes were measured three times per week using calipers [28,29]. The tumor volumes were calculated with the following formula:tumor volume (cm^3^) = length × width^2^ × 0.5

### 2.2. In Vitro Study

#### 2.2.1. Cell Culture

Cells culture was performed using the same methods described in the in vivo study. Before gas exposure, 0.5 × 10^5^ cells were seeded in 100 mm cell culture dishes (VWR, Leicestershire, UK), followed by incubation at 37 °C and 5% CO_2_.

#### 2.2.2. Sevoflurane Exposure

One day after cell seeding, LLC cells were exposed to sevoflurane for 3 days (sevo group). Cell culture dishes were placed in a 2 L custom-made airtight gas container with inlet and outlet valves, which was maintained in a 37 °C water bath. For sevoflurane delivery, we connected the container inlet to a closed gas delivery system containing a calibrated oxygen flow meter and a sevoflurane vaporizer (Abbott Laboratories, Maidenhead, UK). Outlet gas was monitored (Datex gas monitor, Helsinki, Finland) until a sevoflurane concentration of 2.0% was reached in 2 L/min flow oxygen. Sevoflurane delivery was performed for 1 h per day for 3 days. Subsequently, cells were transferred to a standard cell culture incubator and maintained under standard conditions. Control cells were maintained in the container for 3 days, with a 2 L/min oxygen delivery for 1 h per day. A schematic illustration of the procedures followed in the in vivo study is shown in Figure 2.

#### 2.2.3. Cell Viability Assay

After exposure to 2% sevoflurane, cells were imaged daily for 4 days. At day 4, cell confluency reached about 75%, and cells were counted using a hemocytometer [30].

### 2.3. Statistical Analysis

A normal distribution of data was assessed using the Shapiro-Wilk test. Continuous variables are expressed as mean ± standard deviation (SD) or median [interquartile range]. Statistical significance was evaluated using Mann-Whitney *U*-test or independent *t*-test, according to data normality. The log-rank test using the Kaplan-Meier survival curve was performed. Statistical analyses were performed using R software version 3.6.3 (R Project for Statistical Computing, Vienna, Austria).

## 3. Results

Two mice from the sevo group and one from the control group were excluded from the study because the tumors did not grow after LLC cell injection. There were no statistically significant differences in the survival of mice exposed to sevoflurane compared to control mice (*p* = 0.836). The mean survival of LLC-bearing mice in the sevo group was 28.76 ± 3.75 days, while mice in the control group survived for 29.06 ± 4.45 days (Table 1). As shown in Figure 3, the log-rank test using the Kaplan-Meier survival curve also showed no difference between the two groups (*p* = 0.241). All three tumor-free mice showed rapid weight loss, and their survival was significantly shorter compared to tumor-bearing mice (21.7 ± 3.2 vs. 28.9 ± 4.1; *p* = 0.005).

Tumor size was measured three times per week after LLC cell injection. Tumor size on day 18 was compared between the groups because mice started dying after that day. In mice exposed to sevoflurane, the median tumor size on day 18 was 0.75 cm^3^ (interquartile range, 0.41–1.02 cm^3^), while in control mice, the median tumor size was 0.49 cm^3^ (interquartile range, 0.11–0.79 cm^3^) (Table 1). We found no statistically significant difference in the tumor size between the two groups (*p* = 0.153).

LLC cell invasion in the skin was observed in three mice in the sevo group and four mice in the control group. Survival and tumor size did not differ significantly between mice with and without skin invasion (survival: 27.86 ± 4.67 vs. 29.19 ± 3.93, *p* = 0.449; tumor size: 0.79 ± 0.45 vs. 0.63 ± 0.47, *p* = 0.422).

We also assessed the effects of sevoflurane on LLC proliferation *in vitro*. After cell seeding (day 0), LLC cells were imaged daily (Figure 4a). The microscopic observation revealed that on days 2 and 3, the cell confluency was higher in cells treated with sevoflurane compared to control cells (Figure 4a). Furthermore, cell counting revealed that on day 4, the cell number was significantly higher in the sevo group compared to the control group (9,360,000 ± 497,670 vs. 8,574,800 ± 465,830, *p* = 0.018; Figure 4b).

## 4. Discussion

In current studies, sevoflurane exposure did not affect survival and tumor growth in an LLC xenograft mouse model. However, in vitro it significantly increased the proliferation rate of LLC cells. Interestingly, we observed that three mice that did not develop tumors after LLC cell injection exhibited significant weight loss and shorter survival compared to tumor-bearing mice. It is likely that in these tumor-free mice, cancer cells might have been injected into the blood vessels. The potential rapid spread of cancer cells through the microcirculation could have contributed to the short survival of these mice. In addition, we observed that LLC cells invaded the skin in seven mice, although the cells were subcutaneously injected into the flank. Nevertheless, no differences in survival and tumor growth were observed in these mice compared to mice without LLC cell skin invasion.

There is inconsistent evidence concerning effect of inhalational anesthetic agents on proliferation of cancer [2,8,13,14,15,16,17,18,19,20,21,22,23,24,25,26]. There are two main mechanisms by which inhaled anesthetics affect outcomes of cancer. They are supposed to be due to the effect of the anesthetics on the immunomodulation and tumorigenic factors [19,20,21,22,23,24,25,26,27,31]. Cell-mediated immunity acts an important role in preventing spreading and implantation of cancer cells, which are facilitated by the stress response and tissue damage made by surgery [5,6,11,12]. Two in vitro experiments have reported that inhaled anesthetics reduce the activity of natural killer cells, which is important in avoiding proliferations of cancer cells [21,32]. Moreover, quite a few studies have verified that inhaled anesthetics up-regulated level of tumorigenic growth factors, such as hypoxia-inducible factor-1 (HIF-1) and vascular endothelial growth factor (VEGF) [23,24,25,26].

One experiment studied the effects of isoflurane on the expression of tumorigenic markers, such as insulin-like growth factor (IGF-1) in ovarian cancer cells. The experiment reported isoflurane significantly increased expression of IGF-1 and VEGF [13,33]. Laura Benzonana et al. reported that Isoflurane enhanced up-regulation of HIF-1 α and HIF-2α and promoted expression of VEGF A [23]. According to a report published in 2016 by Masae Iwasaki et al., three volatile anesthetics, including sevoflurane, desflurane and isoflurane promoted expression of VEGF-A [26]. In another study that was examined the effect of sevoflurane on glioma stem cells (GSCs) and the mechanisms of action in vitro, expression of VEGF and HIFs were up-regulated by sevoflurane. Therefore, it was reported as sevoflurane could enhance the increase of human GSCs due to upregulate level of HIFs in vitro [25].

A. Buckley et al. investigated the NK cell cytotoxicity depending on anesthetic methods. In this study, serum of breast cancer patients underwent surgical excision were classified according to the anesthesia method. NK cell cytotoxicity was significantly higher in serum of patients who were performed propofol–paravertebral block than in serum of the patients who administrated sevoflurane–opioids [21]. In a in vivo experiment, General anesthesia using isoflurane following by induction with propofol reduced the NK cytotoxic activity of peripheral blood lymphocytes in dogs [27].

LLC was separated from spontaneous epidermal cancer of the lungs of a mouse in 1951, and has since been used to set up animal models with tumor [34]. It is a rapidly growing tumor and a very malignant type of epidermoid carcinoma [35,36]. LLC is a murine tumor, and we have experimented with LCC cells because the tumor grows well in a mouse and is easy to experiment without using a special mouse such as a nude mouse with reduced immunity. However, because LLC cells are cancer cells that grow only in mice and the proliferation rate may be faster than other cancer cells [34,35,36], which may affect the experimental results.

In the present study, we investigated whether the effects of sevoflurane on the proliferation of LLC cells in vitro and in vivo. We found that although sevoflurane exposure enhanced LLC cell proliferation in vitro, it did not affect tumor growth or survival in LLC-bearing mice. These results suggest that in vitro studies on the effects of anesthetics on cancer may differ from those of in vivo or clinical studies. In vivo study, the in vitro effect of 2% sevoflurane may not be properly expressed because the immune system and various environments of the living mice affect it, or it may be because the number of final enrolled mice in vivo study was too small (*n* = 34). Although 2% sevoflurane significantly increased the in vitro cancer cell density, the difference was only 9.2%. It would be difficult to detect a similar change in either animal survival or tumor size.

The limitations of this study are as follows. First, it was not possible to reveal the proportional relationship at other concentrations by using only the 2% sevoflurane concentration. At concentrations of 4% or more, the respiration rate and heart rate of mice were significantly decreased. As a result of concern about bias due to hypoxia or acidosis, sevoflurane at a high concentration could not be applied. To this end, there is a difficulty in performing positive pressure controlled breathing in mice and adding arterial blood gas tests. The authors generally use sevoflurane to anesthetize patients with a sufficient dose of remifentanil, which is continuously administered intravenously, so only a small amount of sevoflurane (1–2%) is used except during periods of strong stimulation or elevated blood pressure. The concentration of sevoflurane, which is most commonly used in clinical practice, is about 1–2%. Second, the size or weight of the cancer tissue was not directly measured, but this was to record and compare the survival period. It is now attempting to measure the occurrence and size of tumor more accurately using ultrasonic waves. The use of 100% oxygen rather than propofol as a control is the third limitation of this study. If there had been a comparison with propofol, it would have been a more valuable study. Finally, it is also a weak point of this study that experiments with animal lung cancer cell, not human cancer cell, and with a non-surgical model rather than a surgical model are a setting that is somewhat far from the actual clinical situation.

## 5. Conclusions

In our experiments, 2% sevoflurane was found to activate proliferation of LLC cells in vitro study. But there was no significant effect on outcomes such as survival or tumor size in vivo study. These results suggest that in vitro studies on the effects of anesthetics on cancer may differ from those of in vivo or clinical studies. Further studies are needed on the effects of sevoflurane on human cancer cells, their mechanism of action, and the effect of sevoflurane at different concentrations.

## Figures and Tables

**Figure 1 medicina-57-00045-f001:**
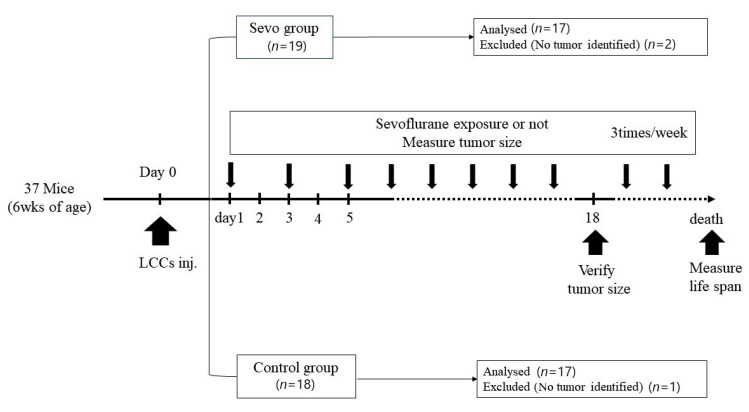
Schematic illustration of the procedures followed in vivo study. LCC, Lewis carcinoma cell; sevo, sevoflurane.

**Figure 2 medicina-57-00045-f002:**
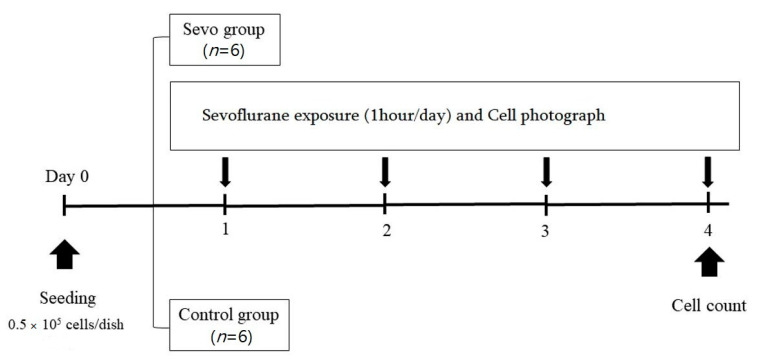
Schematic illustration of the procedures followed in vitro study. Sevo, sevoflurane.

**Figure 3 medicina-57-00045-f003:**
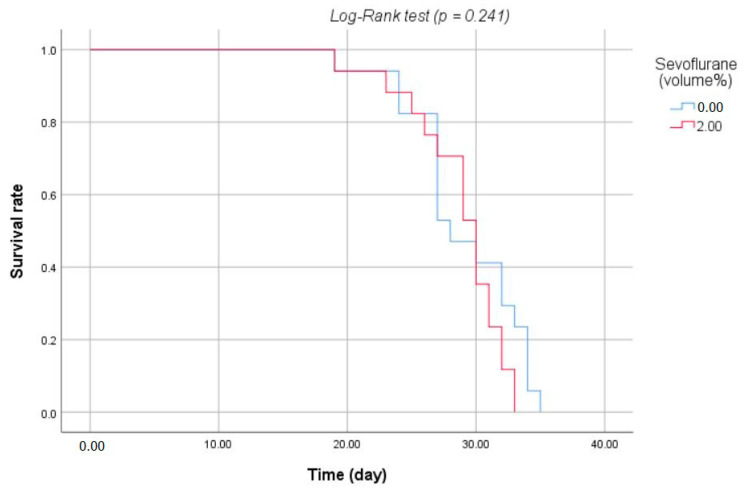
Kaplan-Meier survival curve with a log-rank test after xenograft.

**Figure 4 medicina-57-00045-f004:**
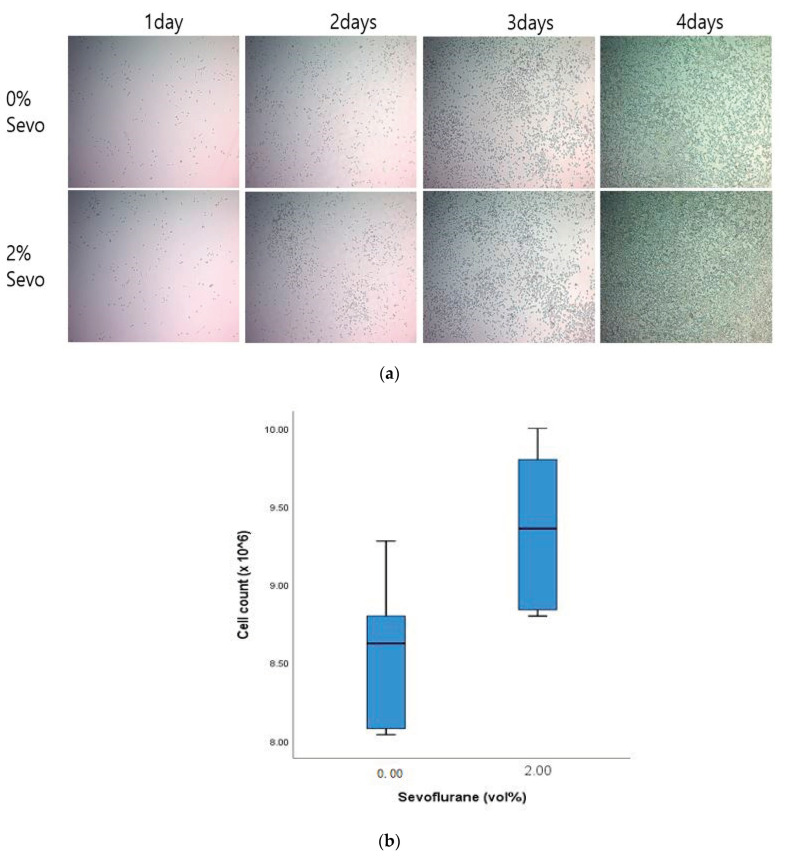
Representative images of Lewis lung carcinoma cells under the microscope: (**a**) Cell photographs for 4 days after exposure to 2 vol% sevoflurane; (**b**) A comparison of cell count using a hemocytometer on day 4. The cell count was significantly higher in the 2% sevo group to the control (9,360,000. ± 497,670 vs. 8,574,800 ± 465,830, *p* = 0.018). Sevo, sevoflurane.

**Table 1 medicina-57-00045-t001:** Comparison of survival time, tumor size on day 18 and tumor detection time after xenograft.

Variable	Sevo Group (*n* = 17)	Control Group (*n* = 17)	*p-*Value
Weight on day 0 (gram)	21.09 ± 0.94	21.22 ± 0.88	0.660
Weight on day 18 (gram)	22.83 ± 1.68	23.07 ± 1.48	0.659
Tumor size on day 18 (cm^3^)	0.75 [0.40–1.01]	0.48 [0.11–0.78]	0.153
Tumor detection			0.174
Day 12	5 (29.41%)	2 (11.76%)	
Day 13	5 (29.41%)	2 (11.76%)	
Day 15	6 (35.29%)	9 (52.94%)	
Day 17	1 (5.88%)	4 (23.52%)	
Survival time (day)	28.76 ± 3.75	29.05 ± 4.45	0.836

Day means the date after xenograft. Sevo, sevoflurane.

## Data Availability

The raw data are available from the corresponding author on request.

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
