# Peer review of "Effects of Sevoflurane on Lewis Lung Carcinoma Cell Proliferation In Vivo and In Vitro"

_medicina, 2021, doi:10.3390/medicina57010045_

Round 1

Reviewer 1 Report

The study combines in vitro and in vivo approaches to the question of whether sevoflurane affects cancer cell growth. This is a very timely and important question. The authors conclude that sevoflurane promotes the growth of LCC cells in culture, but does not change either the tumor growth or survival in mice. 

Overall, the in vivo study design is weak. The figures suggest that, under control conditions, mouse survival ranges from 18-35 days and the tumor size at 18 days ranges from near zero to 1.75 cm3. This is a large amount of variability. If the effect of sevoflurane was to increase the tumor size by as much as the cell counts increased, 10%, a post-hoc power analysis of the tumor size using 0.5±0.3 as the control and 0.55±0.3 with sevoflurane (alpha = 0.05, beta = 80%), indicates that 560 animals in each group would be needed. Thus, the study may be severely underpowered. 

The concentration of sevoflurane is also a concern. 1 MAC sevoflurane for mice is 2.4%; surgery would not be performed with 2% sevoflurane. 

The current controversy regarding cancer and anesthesia has two aspects. Sevoflurane may be harmful, but propofol may be protective. The fact that this study did not examine propofol lowers its significance.

The figures could be improved. Fig 4 should be referred to in the Methods section, so it should really Fig 2. Figs 2 and 3 are hard to read. There should be more of a distinction between gridlines and the lines within the boxes (I'm not even sure what these represent). I think a better presentation would be to show all data points as dots, and then use a symbol for the median. This would help the reader assess whether the the data are unimodal or bimodal. I would also present the control data on the left side of the graph to make it easier to compare with the cell count figure. The maximum tumor size on the y-axis of Fig 3 should be 2.0, to better compare the groups.

All of the tumor size data (each day) should be presented instead of just 18 days. 

Specific comments.

  1. Abstract line 22. Indicate 4L/min for oxygen.
  2. Abstract line 25. Indicate cells exposed to sevo for 1 hr/day for 3 days.
  3. Methods line 70. Put the information about animals not developing tumors into the Results section. 

Author Response

Dear, reviewer 1.

Thanks for the review, I would like to supplement the content and respond.

The study combines in vitro and in vivo approaches to the question of whether sevoflurane affects cancer cell growth. This is a very timely and important question. The authors conclude that sevoflurane promotes the growth of LCC cells in culture, but does not change either the tumor growth or survival in mice. 

Overall, the in vivo study design is weak. The figures suggest that, under control conditions, mouse survival ranges from 18-35 days and the tumor size at 18 days ranges from near zero to 1.75 cm3. This is a large amount of variability. If the effect of sevoflurane was to increase the tumor size by as much as the cell counts increased, 10%, a post-hoc power analysis of the tumor size using 0.5±0.3 as the control and 0.55±0.3 with sevoflurane (alpha = 0.05, beta = 80%), indicates that 560 animals in each group would be needed. Thus, the study may be severely underpowered. 

-->

First of all, 6 plates were compared in the in vitro study, but in the in vivo study, 17 animals were enrolled in two groups, so a little more number was secured. I think that our study was a more objective comparative study than selecting a mouse whose tumor grew as well as the researcher wanted out of several animals. As you have pointed out, it would be better if we had an in vivo survival study with more mice, but we couldn't test more mice because we were advised to modify the study plan from the facility where the animals were kept as the study proceeded. I hope you understand this.

The concentration of sevoflurane is also a concern. 1 MAC sevoflurane for mice is 2.4%; surgery would not be performed with 2% sevoflurane. 

-->

The authors generally use sevoflurane to anesthetize patients with a sufficient dose of remifentanil, which is continuously administered intravenously, so only a small amount of sevoflurane (1–2%) is used except during periods of strong stimulation or elevated blood pressure. I thought it would be better to plan and conduct animal experiments with a dose of 1 MAC or less, which is the most used in clinical practice. Because of environmental pollution or adverse effects on brain physiology, sevoflurane above 1 MAC is not well used in other centers. The reason for experimenting with sevoflurane at 2% was added in the limitation section.

The current controversy regarding cancer and anesthesia has two aspects. Sevoflurane may be harmful, but propofol may be protective. The fact that this study did not examine propofol lowers its significance.

-->

Studies on other cell lines for Sevoflurane are in progress, and after this study is completed, we plan to proceed with studies on propofol. The use and management of Propofol is very strict in Korea, so we are working hard to qualify facilities and management for this. The lack of a propofol study was also mentioned in the limitation section.

As stated in Limitation:

In vivo study, the in vitro effect of 2% sevoflurane may not be properly expressed because the immune system and various environments of the living mice affect it, or it may be because the number of final enrolled mice in vivo study was too small (n = 34).

The limitations of this study are as follows. First, it was not possible to reveal the proportional relationship at other concentrations by using only the 2% sevoflurane concentration. At concentrations of 4% or more, the respiration rate and heart rate of mice were significantly decreased. As a result of concern about bias due to hypoxia or acidosis, sevoflurane at a high concentration could not be applied. To this end, there is a difficulty in performing positive pressure controlled breathing in mice and adding arterial blood gas tests. The authors generally use sevoflurane to anesthetize patients with a sufficient dose of remifentanil, which is continuously administered intravenously, so only a small amount of sevoflurane (1–2%) is used except during periods of strong stimulation or elevated blood pressure. The concentration of sevoflurane, which is most commonly used in clinical practice, is about 1-2%. Second, the size or weight of the cancer tissue was not directly measured, but this was to record and compare the survival period. It is now attempting to measure the occurrence and size of tumor more accurately using ultrasonic waves. The use of 100% oxygen rather than propofol as a control is the third limitation of this study. If there had been a comparison with propofol, it would have been a more valuable study. Finally, it is also a weak point of this study that experiments with animal lung cancer cell, not human cancer cell, and with a non-surgical model rather than a surgical model are a setting that is somewhat far from the actual clinical situation.

The figures could be improved. Fig 4 should be referred to in the Methods section, so it should really Fig 2. Figs 2 and 3 are hard to read. There should be more of a distinction between gridlines and the lines within the boxes (I'm not even sure what these represent). I think a better presentation would be to show all data points as dots, and then use a symbol for the median. This would help the reader assess whether the the data are unimodal or bimodal. I would also present the control data on the left side of the graph to make it easier to compare with the cell count figure. The maximum tumor size on the y-axis of Fig 3 should be 2.0, to better compare the groups.

-->

Fig. 4 moved to the method part as you pointed out. Figure 2 of the survival period was changed to the Kaplan-Meier curve and created as Figure 3. Figure 4, which compared the size of the tumor on the 18th day, was deleted, and instead, the first time when the tumor was confirmed to occur visually or by palpation was further compared and replaced with Table 1.

All of the tumor size data (each day) should be presented instead of just 18 days. 

-->

The data on the tumor size over the entire period was large and was submitted as a supplement file.

Specific comments.

  1. Abstract line 22. Indicate 4L/min for oxygen.
  2. Abstract line 25. Indicate cells exposed to sevo for 1 hr/day for 3 days.
  3. Methods line 70. Put the information about animals not developing tumors into the Results section. 

-->

I modified everything according to specific comments.

Thank you very much.

Sincerely, Chaeseong Lim, M.D., Ph.D.

24, December, 2020

Reviewer 2 Report

Dear authors, 

Thanks for submitting your work to the journal. You describe in vivo and in vitro experiments aiming to investigate the effects of sevoflurane on proliferation and outcomes of Lewis lung carcinoma (LLC) cells in mice. You exposed the mice to 2% sevoflurane in oxygen (1 h/d, 3 d/week). You exposed cells and assessed proliferation after 3 days.

You did not find any difference in mice but you did find an enhanced LLC cell proliferation in vitro.

You conclude that sevoflurane may promote proliferation of LLC cells in vitro not in vivo, and logically that in vitro studies differ from in vivo or clinical studies.

I have only minor comments.

Studies are well done but interpretation is limited by design. More than one cell line could have been tested, but this is not always possible. This could be mentioned in the limitations section.

For the in vivo part, the interpretation is limited to a non surgical model, which make sense in the context of multiple anaesthesia for different non-surgical procedures in children affected by this cancer. This could be mentioned as the results could have been different in a surgical model.

Days alive could be illustrated on Kaplan-Meier curves with, for example, a log-rank test.

Author Response

Dear, reviewer 2.

I have only minor comments.

Studies are well done but interpretation is limited by design. More than one cell line could have been tested, but this is not always possible. This could be mentioned in the limitations section.

For the in vivo part, the interpretation is limited to a non surgical model, which make sense in the context of multiple anaesthesia for different non-surgical procedures in children affected by this cancer. This could be mentioned as the results could have been different in a surgical model.

-->

As stated in Limitation:

It is also a weak point of this study that experiments with animal lung cancer cell, not human cancer cell, and with a non-surgical model rather than a surgical model are a setting that is somewhat far from the actual clinical situation.

Days alive could be illustrated on Kaplan-Meier curves with, for example, a log-rank test.

-->

As pointed out, Figure 3 is represented by a Kaplan-Meier curve, and the p value of the log-rank test is specified.

Figure 3. Kaplan-Meier survival curve with a log-rank test after xenograft.

Thank you very much.

Sincerely, Chaeseong Lim, M.D, Ph.D.

24, December, 2020.

Round 2

Reviewer 1 Report

The manuscript has been improved especially with the addition of the Kaplan-Meier curve and the discussion of the limitations of the study.

I think that the authors should point out that although sevoflurane increased the in vitro cell density significantly, the change was a modest 10%. It would be difficult to detect a similar change in either animal survival or tumor size. 

The legends to Figs 2 and 3 are crossed out and are not replaced with any other text. I assume that only the Figure number was intended to be changed.

The legends of Figs 2 and 3 should indicate the meaning of the X symbol (mean?) and the horizontal bars within the squares (median?). These horizontal bars should be more clearly distinguishable from the grid lines on the graph.

In the abstract, the results of the in vitro study should be expressed as percent change rather than as cell count numbers. The cell count numbers reflect the conditions under which the experiment were done; they do not have any significance by themselves. 

Author Response

Dear, reviewer.

I think that the authors should point out that although sevoflurane increased the in vitro cell density significantly, the change was a modest 10%. It would be difficult to detect a similar change in either animal survival or tumor size. 

  • I added the contents to the discussion as follows.

Although 2% sevoflurane significantly increased the in vitro cancer cell density, the difference was only 9.2%. It would be difficult to detect a similar change in either animal survival or tumor size.

The legends to Figs 2 and 3 are crossed out and are not replaced with any other text. I assume that only the Figure number was intended to be changed.

  • Figure legends have been modified.

The legends of Figs 2 and 3 should indicate the meaning of the X symbol (mean?) and the horizontal bars within the squares (median?). These horizontal bars should be more clearly distinguishable from the grid lines on the graph.

  • Figures 2 and 3 that had the X symbol were changed to Figure 3 and Table 1, respectively.

In the abstract, the results of the in vitro study should be expressed as percent change rather than as cell count numbers. The cell count numbers reflect the conditions under which the experiment were done; they do not have any significance by themselves. 

  • As pointed out, it would be more accurate to compare the amount of cell growth in a single plate when exposed to sevoflurane compared to the control as a percentage. According to the cell count method I used, I had to count after detaching all the cells, so I did not measure the amount of change every day because I was concerned that the process of attaching after detaching cells every day would act as a bias. Therefore, the average value of 6 plates of each group was calculated only once on the last day after exposure to sevoflurane 3 times. As pointed out, the difference between the two groups was modest at 9.2%, but at first glance, there was a difference between the two groups. It is thought that the inability to conduct more in vivo studies and the low sevoflurane concentration of 2% reduced the significance of the in vivo study. This was described in detail as a limitation in the first revision. Currently, we are conducting experiments with human lung cancer cells other than LLC, and we will apply a method to count the cells every day.

  • The abstract part has been changed as follows.

However, in vitro study, the proliferation of LLC cells exposed to sevoflurane increased by 9.2% compared to the control group (p = 0.018).

Thank you very much.

Sincerely, Chaeseong Lim. M.D., Ph.D.

First, January, 2021.

Reviewer 2 Report

Thanks for this improved version.

Author Response

Dear Reviewer.

Thank you very much.

Happy new year.

Sincerely, Chaeseong Lim, MD, PhD.

First, January, 2021